# Integrating Metabolomics and Genomics to Uncover the Impact of Fermented Total Mixed Ration on Heifer Growth Performance Through Host-Dependent Metabolic Pathways

**DOI:** 10.3390/ani15020173

**Published:** 2025-01-10

**Authors:** Zhenzhen Hu, Minyu Zuo, Shixuan Ding, Yifan Zhong, Mingyuan Xue, Huichao Zheng

**Affiliations:** 1Xianghu Laboratory, Hangzhou 311231, China; 12117005@zju.edu.cn (Z.H.); minyuzuo@webmail.hzau.edu.cn (M.Z.); dingshixuan0330@163.com (S.D.); 2College of Animal Science and Technology, College of Veterinary Medicine, Zhejiang Agricultural and Forestry University, Hangzhou 311300, China; yifanzhong@zafu.edu.cn; 3Institute of Animal Husbandry and Veterinary Science, Zhejiang Academy of Agricultural Sciences, Hangzhou 310021, China

**Keywords:** heifer-growth performance, fermented total mixed ration, genomic analysis, metabolomics, metabolic biomarkers

## Abstract

In recent years, forage resources in many countries have been facing the problems of limited planting area and high cost. Rice and *Zizania latifolia* are widely cultivated, and they produce large amounts of rice straw or sheath and leaves when harvested. However, most of these are discarded or burned, resulting in a waste of resources and environmental pollution. Effective utilization of these by-products is the key to solving the problem of feed shortage in livestock production. The aim of this study was to investigate the effects of fermented total mixed ration containing rice straw and sheath and leaves of *Zizania latifolia* on heifer growth from the perspective of metabolism and host genetics. Feeding fermented total mixed ration containing 35% rice straw and 31% sheath and leaves of *Zizania latifolia* could improve the nutrient intake and antioxidant capacity of heifers. And the key metabolites involved in energy metabolism and oxidative balance can be used as candidate markers for heifers’ average daily gain. This integrated omics approach highlights the potential to increase livestock productivity and promote sustainable agricultural practices.

## 1. Introduction

Over the years, large amounts of crop by-products have been incinerated or discarded, and a very low proportion have been used as animal feed, resulting in significant environmental pollution and a waste of resources [1]. When considering the use of crop by-products as animal feed, researchers also found that direct feeding had drawbacks, such as low nutritional value and poor palatability [2]. At present, fermented feeds have garnered increasing attention due to their superior nutritional properties, digestibility, palatability, and safety compared to traditional feeds [3]. These feeds are produced by introducing natural or exogenous microorganisms into the feed matrix under controlled conditions [4]. Through fermentation, antinutritional factors are decomposed or converted into non-toxic substances, while the concentration of beneficial probiotics, enzymes, and metabolites is raised [5]. Therefore, the widespread use of fermented feeds can alleviate pressure on feed resources, promote the utilization of agricultural by-products, and address key challenges in the development of the feed industry and livestock agriculture.

Rice, which is a primary food source for more than half of the world’s population, produces large quantities of rice straw. *Zizania latifolia*, as an important aquatic vegetable, is widely cultivated and harvested to produce a large number of sheaths and leaves. Thus, effective management of these agricultural by-products is essential to address feed shortages in livestock production [6]. Research has shown that treating rice straw with Aspergillus terreus improves digestibility and reduces methane production in goats [7], while mixing rice straw with whole sugar beet silage improves nutritional value and performance in dairy cows [8]. Despite these findings, the effects of the use of rice straw and sheath and leaves of *Zizania latifolia* as fermented feed on heifers remains unclear.

Fermented total mixed ration (FTMR) is encouraged to use low-cost feed substitutes for fermentation. Different fermentation methods or different substrates often change feed composition, such as fiber components, crude protein, organic acids, etc., and then change nutrient digestibility, which may affect animal metabolism. As the final products of gene regulation and protein interactions, metabolites deliver intricate biological information about life processes [4]. In recent years, metabolomics has become a more and more widely used research method. Metabolomics can uncover comprehensive metabolic profile shifts in biological phenotypes, and employing metabolomics to examine the metabolome provides insights into its various influences. Researchers can discover new biomarkers by analyzing changes in the expression profile of metabolites to gain further insight into important metabolic pathways associated with the studied trait through the identification of biomarkers to select high-trait animals [9,10].

Growth and metabolism are critical to the health of heifers and their subsequent performance. Therefore, the aim of this study was to assess the alterations in metabolic profile under host genetic regulation of heifers fed a FTMR containing rice straw and sheath and leaves of *Zizania latifolia* and its effects on cattle growth. By clarifying the application of these two common crop by-products to efficient feed utilization of heifers, it can be demonstrated how precision nutrition and genetic analysis can optimize cattle growth by identifying the specific impacts of different feed components and key metabolic biomarkers. These integrative strategies seek to enhance livestock productivity and promote sustainable agricultural practices.

## 2. Materials and Methods

### 2.1. Animal Ethics

The experiment was conducted at the Ningbo Milk Group Company Limited (Ningbo, China). And this research adhered to Chinese guidelines for animal welfare and was conducted under the experimental protocols (No. 2023ZAASLA043) sanctioned by the Animal Care Committee of Zhejiang Academy of Agricultural Sciences (Hangzhou, China).

### 2.2. Animals, Experimental Design, and Diets

The study consisted of 157 heifers from Ningbo Milk Group Co. (age in days = 234.15 ± 18.23, body weight = 264.45 ± 30.90). Cattle were managed in a consistent naturally ventilated barn for 2 weeks before treatment to acclimatize to the environment, facility, and feeding patterns. And then 75 cattle (age in days = 233.02 ± 16.13, body weight = 262.11 ± 21.76) were randomly divided into four groups with similar age in days and body weight: 18 cattle for feeding FTMR containing 21% rice straw (LSF), 18 cattle for feeding FTMR containing 28% rice straw (MSF), 18 cattle for feeding FTMR containing 35% rice straw (HSF), and 21 cattle for feeding FTMR containing 31% sheath and leaves of *Zizania latifolia* (ZF), and the trial period was 2 months (Figure 1). The proportion of FTMR rice straw/ sheath and leaves of *Zizania latifolia* added to the four treatment groups of breeding cattle, and the nutrient levels are shown in the feed ingredient table (Table 1). During the experiment, the average temperature of the barn was 30.0 °C and the average humidity was 76.5%. Throughout the whole period, all animals had free access to drinking water and feed.

### 2.3. Measurement of Dry Matter Intake and Body Weight

Throughout the trial, the FTMR intake of each treatment group was tracked on a daily basis, and the dry matter intake of each treatment group was calculated. The cattle were, respectively, weighed at the beginning and end of the trial period for 3 consecutive days before morning feeding. The average daily gain (ADG) in body weight was calculated according to the following formulae [11]:$$average\;daily\;weight\;gain\;\left(kg/d\right)=\frac{final\;body\;weight\;\left(kg\right)-initial\;body\;weight\;\left(kg\right)}{experimental\;days\;\left(d\right)}$$

And the collection of data on birth weight of 75 cows was undertaken as underlying data to serve as an approximate reference for weight gain during the trial.

### 2.4. Sample Collection

Dozens of hairs with intact follicles from all 157 cattle were collected from the tail of each heifer separately, were packed in clean self-sealing bags, sealed and labeled, and frozen at −20 °C for subsequent genotyping [12]. At the end of the experiment, blood samples were obtained from the jugular veins of each heifer while they were fasting and collected into 5 mL pro-coagulation tubes [13]. After centrifuging at 3000× *g* for 15 min, the serum samples were collected and kept at −80 °C until analysis. A total of 50 mL of urine was collected by vulvar stimulation in 50 mL centrifuge tubes, which were divided into 3 cryopreservation tubes. Immediately after quenching in liquid nitrogen, the samples were stored at −80 °C in a deep freezer [14].

### 2.5. Biochemical Index Measurements

All necessary experimental equipment was prepared, including a centrifuge, pipette, test tubes, cuvettes, a constant temperature water bath, and an automatic biochemical analyzer. It was ensured that the laboratory environment was clean, with appropriate temperature and humidity. Samples were collected, and biochemical parameters in serum and urine—such as glucose, blood urea nitrogen, triglycerides, total protein, albumin, globulin, albumin/globulin (A/G) ratio, uric acid, and urine urea nitrogen—were analyzed utilizing colorimetric commercial kits provided by Ningbo Medical System Biotechnology Co., Ltd., located in Ningbo, China, alongside an Auto Analyzer 7020 instrument from Hitachi High-Technologies Corp., Tokyo, Japan. Absorbance values were read using an automatic biochemical analyzer (7170A, HITACHI, Tokyo, Japan), and data were recorded [15]. The concentration of each indicator was calculated according to the formula or standard curve provided in the kit.

### 2.6. GC-TOF-MS-Based Untargeted Metabolomics Analysis

#### 2.6.1. Metabolites Extraction

A comparison was conducted between the effects of the fermented total mixed diets (LSF, MSF, HSF, ZF) with different proportions of added rice straw on the urinary metabolome of Chinese Holstein breeding cattle. Approximately 100 μL of the sample was transferred to an EP tube, followed by the addition of 20 μL of urease (80 mg/mL). The mixture was then incubated for 1 h. Next, 360 μL of a methanol-ribose solution (with a ratio of 350:10) was added and vortexed. After 10 min of ultrasound treatment in an ice water bath, the samples were centrifuged at 4 °C and 12,000 RPM for 15 min. Carefully transferring the resulting 180 μL supernatant into a 1.5 mL EP tube, 60 μL of each sample was combined to create quality control (QC) samples. The extract was dried using a vacuum concentrator. Subsequently, 80 μL of methoxyamination hydrochloride (20 mg/mL in pyridine) was incorporated and maintained at 80 °C for 30 min. The samples were then derivatized with 100 μL of BSTFA reagent (1% TMCS, *v*/*v*) at 70 °C for 1.5 h. Finally, once the temperature was gradually reduced to room level, 5 μL of FAMEs (dissolved in chloroform) was introduced to the QC sample. Every sample underwent analysis via gas chromatography paired with a time-of-flight mass spectrometer (GC-TOF-MS).

#### 2.6.2. GC–TOF/MS Analysis

An Agilent 7890 gas chromatograph (Agilent Technologies, Santa Clara, CA, USA) paired with a time-of-flight mass spectrometer (Sequenom Company, San Diego, CA, USA) was used for GC-TOF-MS analysis, featuring a DB-5MS capillary column. A 1 μL sample aliquot was injected in splitless mode. Helium was utilized as the carrier gas, with a purge flow at the front inlet of 3 mL/min and a column gas flow rate of 1 mL/min. The temperature started at 50 °C for 1 min and was then increased to 310 °C at a rate of 10 °C per minute and was kept for 8 min at 310 °C. The injection, transfer line, and ion source temperatures were 280, 280, and 250 °C, respectively. With an energy of −70 eV in electron-impact mode, mass spectrometry data were obtained in full-scan mode over an *m*/*z* range of 50 to 500, at a speed of 12.5 spectra per second, with a solvent delay of 6.27 min.

#### 2.6.3. Data Analysis

Chroma TOF software (V 4.3x, LECO) (LECO Corporation, St. Joseph, MI, USA) was used to complete the raw data analysis, which involved peak extraction, baseline correction, deconvolution, alignment, and integration. Metabolite identification was carried out by matching the mass spectra and retention indices with the LECO-Fiehn Rtx5 database. Peaks found in less than 50% of QC samples or having a relative standard deviation (RSD) exceeding 30% in QC samples were then excluded from the dataset.

SIMCA16.0.2 (V16.0.2, Sartorius Stedim Data Analytics AB, Umea, Sweden) was used to input the data for principal component analysis (PCA) and orthogonal projections for latent structure–discriminate analysis (OPLS–DA). Using the R package MetaboAnalystR, differential analysis and volcano plotting were carried out to discover differential metabolites between various groups, based on VIP > 1 and *p* < 0.05. In addition, the Kyoto Encyclopedia of Genes and Genomes (KEGG), (http://www.kegg.jp/kegg/pathway.html, accessed on 15 January 2024) was used to search for the pathways of the differential metabolites. MetaboAnalyst 6.0 (http://www.metaboanalyst.ca/, accessed on 28 January 2024) was used for pathway enrichment analysis. An impact score above 0.1 was employed to identify the significantly enriched metabolic pathways. Furthermore, we chose the top 10 metabolites from the differential metabolites between the groups for correlation analysis, and the overall metabolites among the groups were shown in the chord chart.

### 2.7. Correlation Analysis and Random Forest

Using the R package, Hmisc, we conducted a correlation analysis to examine the relationship between metabolites and ADG, identifying metabolites with an R > 0.3 as significantly correlated. Random forest modeling (R package “randomForest,” version 4.6–14) was used to identify whether specific urine metabolites contributing to the ADG of body weight could potentially indicate heifer growth. The technique of machine learning addresses nonlinear relationships and dependencies among all metabolites. Each metabolite from the predictors received a score showing its importance (%IncMSE) based on the rise in error when it was removed. In random forest modeling, 70% of the data is used as a “training” set through random sampling with replacement, while the remaining “out-of-bag” samples are used for validation of the chosen metabolites. We determined the optimal predictive model by maximizing the area under the curve (AUC) by using the AUC-RF-algorithm.

### 2.8. Genome-Wide Association Analysis

Using the Geneseek Genomic profiler (GGP) NEOGEN 100K SNP Chip from the NEOGEN company (Neogen, Lincoln, NE, USA), the DNA of hair follicles was extracted and genotyped. Simultaneously, Plink software (v1.90, Cambridge, MA, USA) was utilized for quality control (QC) to discard markers failing to meet these standards: (1) a call rate for individual SNP genotypes under 98%, (2) the minimum allele frequency (MAF) of SNP being ≥0.05, (3) significant deviation from Hardy–Weinberg equilibrium ≥ 1.0 × 10^−4^, and (4) the heterozygosity rate > ±3 SD. After quality control, there remained 72 cows and 72,953 SNP variants for further association analysis. Using the Bonferroni correction method, we recognized the importance of the threshold value in selecting significant SNPs. The genome-wide threshold for significance was 6.85 × 10^−7^ (0.05/72,953). Manhattan plots were generated using the R qqman package.

### 2.9. Statistical Analysis

Normal distribution was tested by the Shapiro–Wilk test, while homogeneity of variance was checked with the Levene test. The experimental data all were consistent with a normal distribution. And One-way ANOVA followed by Tukey’s post-hoc test was conducted to analyze the significance of data differences using R software (version 4.2.0). The Bray–Curis distance metric was calculated to assess the variability in urinary metabolomic profiles among the samples. To evaluate the significance of structural differences in urinary metabolomics, permutation multivariate analysis of variance (PERMANOVA) was employed. Additionally, Spearman correlation analysis for differential metabolites was performed using the Hmisc package in R (version 4.6.0, http://www.R-project.org/, accessed on 22 May 2024).

## 3. Results

### 3.1. Feed Intake and Growth Performance

Diet price is presented in Table 1 as the basic data. As the proportion of rice straw in FTMR increased from 21% to 35%, the feed cost per ton gradually decreased, and the ZF group was cheaper than the three groups containing rice straw in FTMR. As shown in Table 2, the group feed intake across all groups showed fluctuations but was generally similar. Compared to the LSF group, the MSF group, HSF group and ZF group were lower by 3.8%, and 1.2% and higher by 3.8%, respectively. The ZF group was higher by 5.1% than the HSF group. Regarding ADG, there was no notable difference among the groups (*p* > 0.05).

### 3.2. Plasma and Urine Parameters

The results of the biochemical indexes are shown in Figure 2. Compared to the LSF group, the HSF group exhibited a significantly higher plasma glutamic acid (GLU) level (*p* < 0.05). Additionally, blood urea nitrogen (BUN) levels were significantly higher in both the HSF and ZF groups (*p* < 0.05), while A/G in the MSF group was significantly lower (*p* < 0.05). There were no significant differences in triglyceride (TG), total protein (TP), albumin (ALB), and globulin (GLOB) among all groups. Uric acid in the ZF group was significantly lower than that in the LSF group (*p* < 0.05), while urine urea nitrogen (UUN) was significantly higher (*p* < 0.05).

### 3.3. Urine Metabolome

Metabolomics profiling of urine samples was performed and the differences in urine metabolomic features among four treatment groups are shown in Figure 3. The PCoA result was shown in Figure 3A, and no significant separation was found among the groups. Pairwise-adonis analysis showed that there was a significant difference between the HSF group and the LSF group (*p* < 0.01), indicating that the metabolite patterns between the two groups were different. Differential metabolites were analyzed among the groups, as shown in Figure 3B. We found 50 different metabolites between the HSF group and the LSF group, of which 33 were up-regulated and 17 were down-regulated in the HSF group. There were 107 different metabolites between the ZF group and LSF group, of which 86 were up-regulated and 21 down-regulated in the ZF group. And there was a total of 39 different metabolites including 31 up-regulated and 8 down-regulated in the MSF compared with the LSF group. And 56 different metabolites were identified between the HSF group and ZF group, among which 39 were up-regulated and 17 were down-regulated in the HSF group.

Using the KEGG database, pathway enrichment analysis was carried out on differential metabolites, as shown in Figure 3C,D. According to the pathway impact > 0.1, we found that the LSF–HSF groups can significantly enrich the riboflavin pathway, and the differential metabolites included p-benzoquinone and ribitol. And the LSF–MSF groups were significantly enriched into sulfur metabolism pathways, with differential metabolites including sulfuric acid, o-acetylserine, and l-homoserine. In addition, the glutathione metabolism pathway was significantly enriched into the ZF–HSF group, where differential metabolites included ascorbate, l-glutamic acid, glycine, and glutathione—H_2_O.

In order to further understand the specific changes of differential metabolites between groups, we selected the top 10 differential metabolites of fold change among the groups. And we found that compared with the LSF group, three metabolites in the HSF group were up-regulated and seven metabolites were down-regulated (Figure 4A), and in group ZF, five were significantly increased and five were significantly decreased (Figure 4B). The levels of two metabolites were significantly increased, while the levels of eight metabolites were significantly decreased in the MSF group (Figure 4C). At the same time, compared with the ZF group, the HSF group exhibited a significant rise in three differential metabolites and a significant decline in eight (Figure 4D). In addition, the chord plots and correlation analysis revealed varying degrees of correlation between these metabolites (Figure 4E–H).

### 3.4. Individual Variability of Urine Metabolites Affecting ADG

In order to explore the effects of metabolites on ADG, correlation analysis was first conducted between all metabolites and ADG. A total of 76 metabolites were found to be significantly correlated (*p* < 0.05), of which 28 were metabolites with R > 0.3, 18 were metabolites with 0.3 > R > 0.25, and 30 were metabolites with 0.25 > R > 0.2 (Figure 5A). Then, the relative importance of R > 0.25 metabolites was studied by random forest regression. Figure 5B showed the IncMSE value of metabolites representing the importance rank. Then, we conducted AUC analysis of metabolites in order of importance (Figure 5C). Curve analysis showed that the AUC reached its maximum when n = 12, when AUC = 0.7736 (Figure 5D). The 12 metabolites included galactonic acid, 1-hydroxyanthraquinone, fluorene, phloretin, xylitol, n-acetyl-l-leucine, citramalic acid, citraconic acid, 4-hydroxybutyrate, m-cresol, and 3-ureidopropionate.

### 3.5. Genetic Affection on Host Metabolism

The genetic data are available on the Genome Variation Map (GVM) in the China National Center for Bioinformation under the accession number of GVM000829 (https://bigd.big.ac.cn/gvm/getProjectDetail?Project=GVM000829, accessed on 6 August 2024). Weight-related indexes, including birth weight, daily gain from birth to the experimental period, daily gain during the 60-day experimental period, and coefficient of variation of metabolites were analyzed among all groups (Figure 6). It was found that the coefficient of variation of metabolites was higher than that of weight-related indexes on the whole, suggesting that metabolites varied greatly among individuals. Therefore, we conducted GWAS analysis from 75 cattle on metabolites and SNP sites with R > 0.3. Among the biochemical indicators, GLU and GLOB in plasma and UUN in urine had significant associated SNPs (Figure 7A–C). Among the 11 key differential metabolites obtained from the difference analysis between different groups and KEGG enrichment analysis, 2 metabolites were significantly associated with SNP sites (Figure 7D,E). In addition, among the 12 core metabolites screened by the ADG correlation analysis and machine learning, 4 metabolites were significantly associated with SNPS, which were m-cresol, phloretin, 4-hydroxybutyrate, and xylitol (Figure 7F–I).

## 4. Discussion

Improving the feed value of crop by-products not only reduces the environmental pollution caused by burning or discarding, but also meets the urgent need of feed supply for ruminants [16]. Biological fermentation technology can decompose complex organic matter in crop by-products into small molecular nutrients, which is easy for animals to digest and absorb, and which improves its utilization rate [17]. However, the use of rice straw and sheath and leaves of *Zizania latifolia* as common crop by-products in heifers has been limited. This study aimed to mine the core metabolic markers related to growth performance through metabolome and genomic information to determine the application effects and potential mechanisms of these two common crop by-products in the fermented diet for heifers, so as to realize the feed utilization of rice straw or sheath and leaves of *Zizania latifolia* in heifers.

Rice straw is a common crop straw, which contains a large number of complex lignocellulosic polymers, pectin, silica, and wax, which prevent microorganisms and enzymes from approaching cellulose and hemicellulose, and which thus affects the digestion and absorption of animals [18]. Studies have shown that the nutritional value of fermented straw is significantly improved compared to unfermented straw [19,20]. In this study, it was found that the NDF and ADF contents in FTMR decreased with the increase of rice straw proportion, and the silage fermentation quality of 35% straw proportion was the best, which improved the feed quality. In addition, the levels of NDF in fermented diets supplemented with 31% sheath and leaves of *Zizania latifolia* were higher than those in rice straw fermented diets. Moreover, the reasonable adding ratio effectively helped to reduce the feed cost. At present, there are few studies on the use of sheath and leaves of *Zizania latifolia* in fermented diets.

Heifers are an important stage of dairy cattle breeding, and the nutrient composition of their diet is an important factor that affect the growth performance and metabolism level of heifers [21]. Differences in metabolites between heifers in different diet groups were assessed to identify representative metabolites related to the diet. In this study, with the increase of the proportion of rice straw, the levels of blood glucose and urea nitrogen in the jugular vein were continuously increased, and the levels of uric acid and urea nitrogen in urine continuously increased, indicating that the high proportion of rice straw fermented feed may improve the level of blood glucose and non-protein nitrogen intake of heifers [22]. Recently, we have established the potential causal relationship between microbial metabolites and phenotypes by integrating host genetics and microbial metabolome data [23]. In this study, we also combined the host genome to analyze the association between metabolites and phenotypes. Using GWAS analysis, we found that the presence of SNPs was significantly associated with glucose and urea nitrogen. This suggests that these two metabolites are key metabolites subject to genetic regulation. After digestion of rice straw fermented feed in the rumen and intestine, nutrients are transported to the liver for further metabolism, and the metabolic level may be related to the feed silage effect [24]. The changes of biochemical indexes from the jugular vein and urine usually reflect the level of liver and kidney metabolism. Urea nitrogen levels in blood and urine are the end products of protein and amino acid metabolism, and the increase of its level indicates that the body may have high amino acid levels [25]. Urea nitrogen and uric acid, both products of protein metabolism, are excreted from the body through the kidneys, and their elevated levels indicate a strong metabolic function in the kidneys. At the same time, elevated liver gluconeogenesis levels can increase blood glucose levels [26]. Metabolomic analysis between groups identified nine metabolites with significant differences, including o-acetylserine, sulfuric acid, glycine, p-benzoquinone, glutathione-H_2_O, ascorbate, l-glutamic acid, and l-homoserine. Among them, SNPs were significantly associated with l-homoserine and o-acetylserine. Their respective functions were as follows: O-acetylserine can promote the formation of cysteine from serine and acetyl-CoA by serine acetyltransferase. It was also capable of producing glutathione by inducing the activation of adenosine 5′-phosphosulphate reductase activity and sulfur flux [27]. Glutathione was a tripeptide with important antioxidant functions. It was closely related to cellular REDOX homeostasis and related signaling. Together with glutathione-dependent enzymes, it controlled the levels of specific oxides, such as H_2_O_2_ and reactive aldehydes [28]. In addition to being the main form of the trace element vitamin C in the body, ascorbate was a very effective primary scavenger of reactive oxygen species. It and glutathione were the two main water-soluble antioxidants in the body [29]. Glycine can promote the oxidation of glutathione to glutathione disulfide and change the REDOX level [30]. L-glutamic acid can slow down the decrease of antioxidant enzyme activity and increase of lipid peroxidation products in poisoned rats, so it also played a certain antioxidant role [31]. It has been reported that l-homoserine is a non-essential chiral amino acid that can be converted into many chiral compounds. Therefore, l-homoserine was an important biosynthetic intermediate that may enhance neuronal and muscle development in vivo, and it also enhanced the expression of matriglycan to regulate the cell cycle [32]. However, sulfuric acid and p-benzoquinone have been widely studied in vitro, such as in relation to changing the composition of feed, but there are few reports on their physiological effects in animals [33,34]. Wang et al. studied metabolic profiling of dairy cows after feeding straw to find nutritional or genetic regulatory targets to improve cow performance, and they also found relevant biomarkers, such as hippuric acid [35]. This is an effective research tool to analyze the mechanism of animal phenotypic changes by analyzing and identifying metabolite markers [36,37]. In this study, through biomarker analysis, we found that the above key metabolites mainly functioned around glutathione and its antioxidant function, and their levels were higher in the HSF and ZF groups, suggesting that FTMR with a higher proportion of rice straw and sheath and leaves of *Zizania latifolia* may have a stronger antioxidant capacity centered on glutathione metabolism.

However, there was no significant difference in ADG between the groups in this study. It was found that, on the whole, metabolites exhibited a higher coefficient of variation than weight-related parameters [38]. Therefore, it was very likely that the influence of metabolites on ADG was covered up by the large individual differences of the metabolites. In this study, we used correlation analysis and random forest analysis, and found that 28 metabolites were significantly positively correlated with ADG, and with the increase of the number of metabolites, the AUC was the largest at n = 12. They were galactonic acid, 1-hydroxyanthraquinone, fluorene, phloretin, xylitol, n-acetyl-l-leucine, citramalic acid, citraconic acid, 4-hydroxybutyrate, m-cresol, 3-ureidopropionate, and ribitol. Through GWAS analysis, it was determined that there were four metabolites with significant SNPs, which were m-cresol, phloretin, 4-hydroxybutyrate, and xylitol. We speculated that these four metabolites had potential as biomarkers of “high ADG performance” and were key metabolites under genetic regulation. The functions of these metabolites were as follows: galactonic acid was mainly involved in galactose metabolism and the Tricarboxylic Acid Cycle (TCA) cycle, and was a galactose downstream metabolite [39]. 1-Hydroxyanthraquinone was an anthraquinone compound with anti-inflammatory effects in vivo, as when it played a key anti-inflammatory role in mastitis mice [40]. There was a negative correlation between fluorene and DNA methylation [41]. Phloretin was a planar lipophilic polyphenol that can attack many molecular targets and regulate different intracellular signaling pathways, such as being able to resist oxidation, anti-proliferation, pro-apoptosis, anti-metastasis, and anti-angiogenic activities [42]. Xylitol, a 5-carbon sugar alcohol (polyol), was a low-abundance intermediate byproduct of the glucuronic acid pathway in glucose metabolism, which can enhance platelet reactivity and thrombotic potential in vivo [43]. N-acetyl-l-leucine normalized glucose and glutamate metabolism and increased autophagy and superoxide dismutase levels. It can improve glucose metabolism, mitochondrial energy metabolism, and antioxidant processes [44]. Citramalic acid citrate was the product of the condensation reaction of pyruvate and acetyl-CoA [45]. 4-Hydroxybutyrate was involved in carbon transfer in the TCA cycle, and its concentration increased when succinate was decreased [46]. M-cresol, also known as m-methylphenol, entered the body through the gastrointestinal tract, was metabolized by the liver, and was excreted by the kidney in the form of glucuronic acid and sulfate metabolites. This process may induce oxidative stress and apoptosis and may then cause liver toxicity [47]. 3-Ureidopropionate was a physiological metabolite formed by the hydration of 5,6-dihydrouracil catalyzed by dihydropyrimidinase. It inhibited mitochondrial complex V activity, thereby inhibiting mitochondrial energy metabolism and respiratory chain function. In addition, it was able to cause an increase in reactive oxygen species and promote oxidative stress [48]. Based on the biological functions of these key metabolites, we believed that their functions were mainly focused on energy metabolic processes and REDOX balance. Since the levels of these metabolites were all significantly positively correlated with heifer ADG, the above results suggested that heifers with high ADG had higher levels of energy metabolism and oxidative balance performance centered on these key metabolites.

Multi-omics analysis provided a potential explanation for the effective utilization of fermented feed [37]. The metabolite markers discovered in the results above proposed future evaluation parameters related to the feed utilization of these two crop by-products, promising improvements in heifer performance for better farm management.

## 5. Conclusions

In conclusion, this study showed that increasing the proportion of rice straw in FTMR within a certain range can improve the quality of feed fermentation. And FTMR containing 35% rice straw and 31% sheath and leaves of *Zizania latifolia* may increase the antioxidant level of heifers and improve their ability to resist health risks. In particular, the FTMR with 31% sheath and leaves of *Zizania latifolia* was more cost-effective. In addition, heifers with faster weight gain on these diets were indicated by several metabolites involved in energy metabolism and oxidative balance, which provides a potential direction for precise improvement of heifer productivity in the future.

## Figures and Tables

**Figure 1 animals-15-00173-f001:**
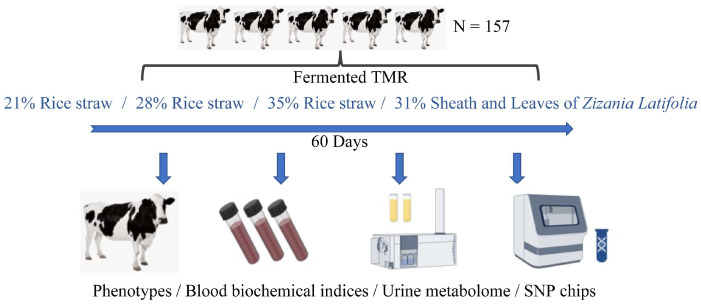
Overview of workflow of the study. A total of 157 dairy heifers were selected and used for genotyping. Four types of fermented TMR were feed for animals, which contained 21%, 28%, 35% rice straw, and 31% sheath and leaves of *Zizania latifolia*, respectively (*n* = 18, 18, 18, and 21 in each group). Phenotypes, blood biochemical indices, and urine metabolome also were performed for experimental animals.

**Figure 2 animals-15-00173-f002:**
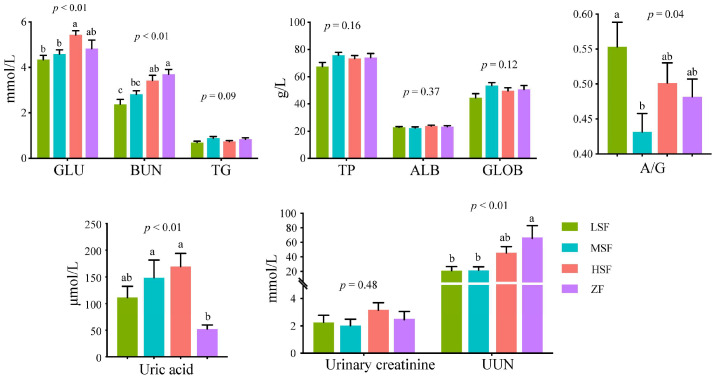
Effects of experimental diets on plasma and urine biochemical parameters. Bar charts showing comparisons among the four groups of biochemical parameters including glucose (GLU), blood urea nitrogen (BUN), triglyceride (TG), total protein (TP), albumin (ALB), globulin (GLOB), albumin/globulin (A/G), uric acid, urinary creatinine, and urine urea nitrogen (UUN). LSF = fermented total mixed ration containing 21% rice straw, MSF = fermented total mixed ration containing 28% rice straw, HSF = fermented total mixed ration containing 35% rice straw, ZF = fermented total mixed ration containing 31% sheath and leaves of *Zizania latifolia*. Different letters (a–c) indicate significant differences among treatments based on a *p* value < 0.05. Each bar represents mean ± SD.

**Figure 3 animals-15-00173-f003:**
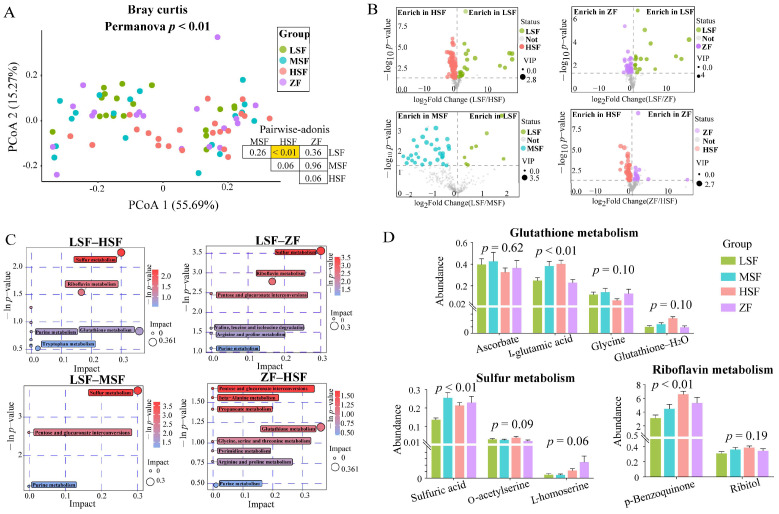
Differences of urine metabolomic features among four treatment groups. (**A**) Principle-coordinate analysis based on Bray–Curtis. (**B**) Volcano plot representing significantly different metabolites between each two groups of LSF vs. HSF, LSF vs. ZF, LSF vs. MSF, ZF vs. HSF. (**C**) Functional enrichment of the significantly different pathways identified by differential metabolites between each two groups of LSF vs. HSF, LSF vs. ZF, LSF vs. MSF, ZF vs. HSF. (**D**) Comparison of metabolites which were used for pathway enrichment in (**C**), among four groups. LSF = fermented total mixed ration containing 21% rice straw, MSF = fermented total mixed ration containing 28% rice straw, HSF = fermented total mixed ration containing 35% rice straw, ZF = fermented total mixed ration containing 31% sheath and leaves of *Zizania latifolia*.

**Figure 4 animals-15-00173-f004:**
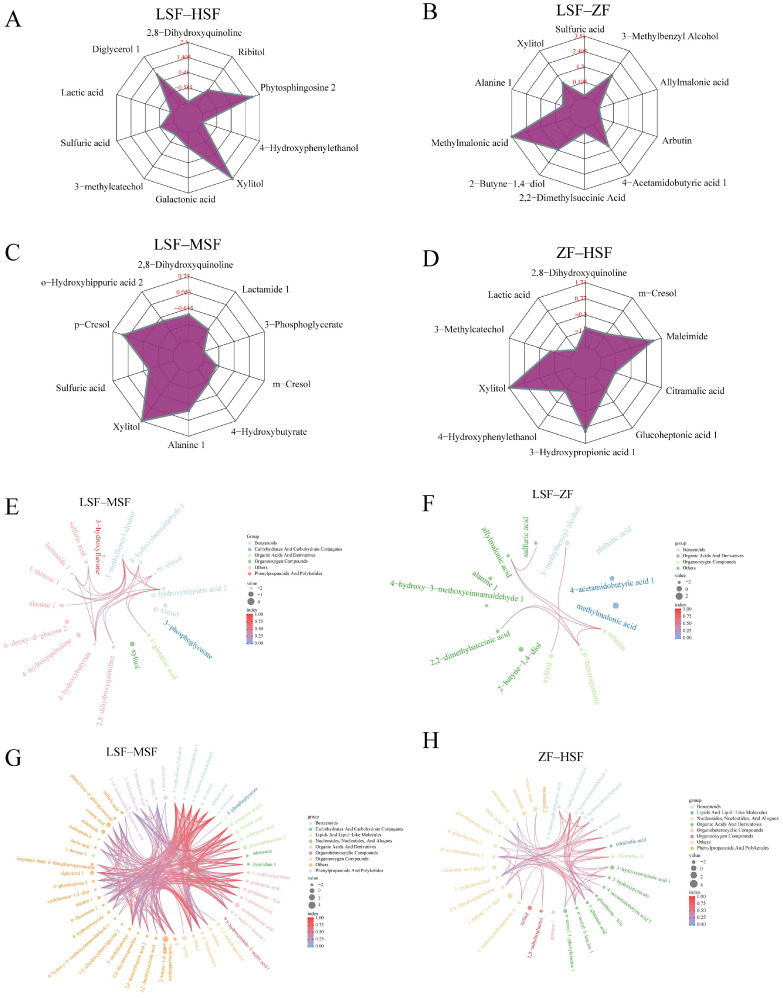
Comparisons and correlations of significantly differential metabolites between each two groups. Radar charts showing the differences in fold changes (log_2_) of significantly differential urine metabolites compared between LSF and HSF (**A**), LSF and ZF (**B**), LSF and MSF (**C**), ZF and HSF (**D**). Correlation networks representing relationships among significantly differential urine metabolites compared between LSF and HSF (**E**), LSF and ZF (**F**), LSF and MSF (**G**), ZF and HSF (**H**). LSF = fermented total mixed ration containing 21% rice straw, MSF = fermented total mixed ration containing 28% rice straw, HSF = fermented total mixed ration containing 35% rice straw, ZF = fermented total mixed ration containing 31% sheath and leaves of *Zizania latifolia*.

**Figure 5 animals-15-00173-f005:**
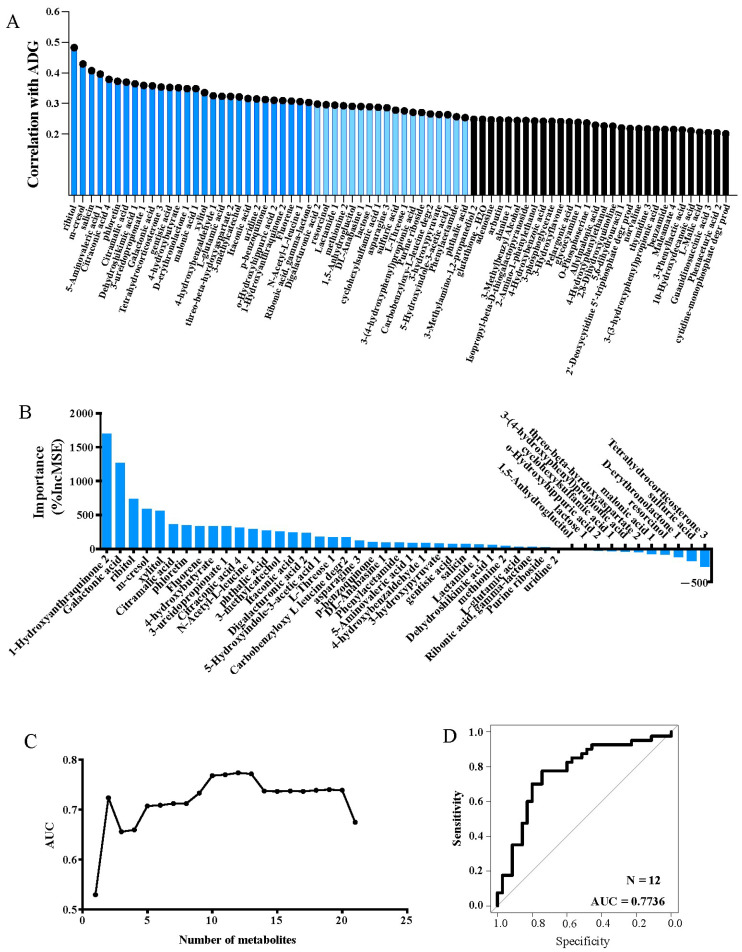
Specific urine metabolites contributing to average daily gain (ADG) of body weight could potentially indicate heifer growth. (**A**) Metabolome features that were significantly correlated with ADG. Dark blue indicates R > 0.3, light blue indicates 0.3 > R > 0.25, and black indicates 0.25 > R > 0.2. (**B**) Prediction of host ADG based on urine metabolites using random forest model. Bar plot shows importance of biomarkers (28 metabolites that were significantly associated with host ADG, R > 0.25) for predicting ADG. (**C**) Number of metabolites that could be used in the model and relative aera under the curve (AUC). (**D**) Receiver operating characteristic (ROC) curve of the performance of the machine-learning model for predicting ADG in the training data set.

**Figure 6 animals-15-00173-f006:**
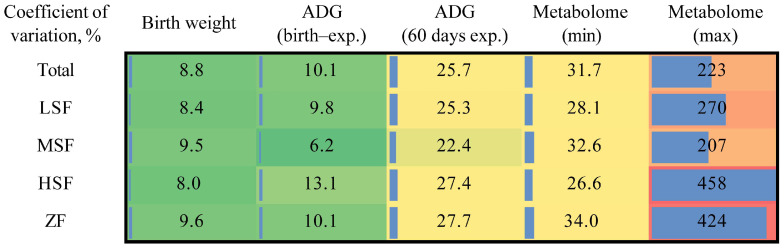
Coefficient of variation of growth phenotypes and metabolome data. Coefficient of variation indices were analyzed based on total animals and each of four groups, respectively. LSF = fermented total mixed ration containing 21% rice straw, MSF = fermented total mixed ration containing 28% rice straw, HSF = fermented total mixed ration containing 35% rice straw, ZF = fermented total mixed ration containing 31% sheath and leaves of *Zizania latifolia*.

**Figure 7 animals-15-00173-f007:**
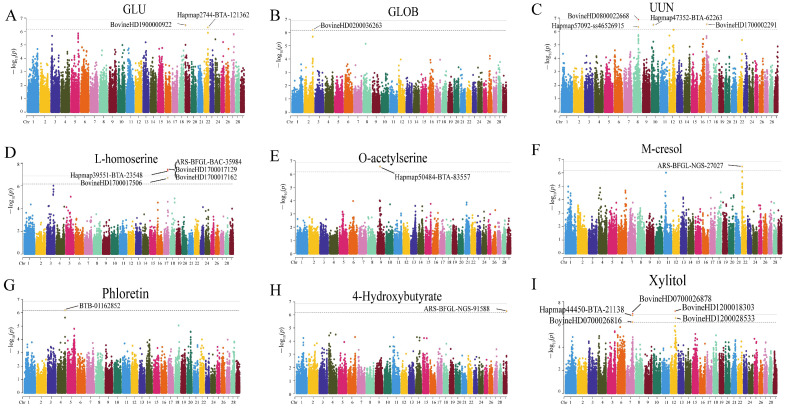
Manhattan plots of single nucleotide polymorphism loci for metabolites. (**A**–**C**) Differential biochemical indices from plasma and urine samples from Figure 2. (**D**,**E**) Differential metabolites between groups from Figure 3D. (**F**–**I**) Core metabolites from Figure 5D.

**Table 1 animals-15-00173-t001:** Ingredients and chemical composition of four experimental diets.

Items	Treatment ^1^			
	LSF	MSF	HSF	ZF
TMR, % of DM ^2^	46.87	46.47	46.19	46.0
Dietary ingredient, % of DM				
Corn	2.4	5.3	9.3	12.4
Soybean meal	6.7	3.8	6.8	3.04
Sprayed corn bran	16.0	16.0	16.0	21.3
Distiller grain	4.6	4.6	4.6	9.37
Whole corn silage	17.0	12.0	7.0	0
Alfalfa hay	10.0	18.0	6.0	0
Oat hay	13.0	0.0	0.0	22.7
TP ^3^	30.0	40.0	50.0	30.8
Stone powder	0.18	0.18	0.18	0.18
Salt	0.18	0.18	0.18	0.18
Chemical composition, % of DM				
Crude protein	14.2	14.2	14.2	14.2
Neutral detergent fiber	49.71	47.53	46.09	52.3
Acid detergent fiber	27.32	26.86	25.46	-
Starch	9.88	10.72	12.83	15.4
Crude ash	7.62	8.34	8.51	6.22
NE_L_ ^4^, Mcal/kg	1.45	1.44	1.43	1.43
Diet price, ¥/t	2152.6	2084.9	2018.2	1749.2

^1^ LSF = fermented total mixed ration containing 21% rice straw, MSF = fermented total mixed ration containing 28% rice straw, HSF = fermented total mixed ration containing 35% rice straw, ZF = fermented total mixed ration containing 31% sheath and leaves of *zizania latifolia.*
^2^ TMR = total mixed ration, DM = dry matter. ^3^ TP was provided by Ningbo Milk Group Company Limited, of which fresh rice straw was 53%, dry straw was 17%, and the total straw was 70% of the dry matter basis. Its dry matter was 39%, and the nutrient composition was calculated on the air dry basis, CP 9.0%, NDF 57.6%, ADF 35.5%, starch 2.9%, Ash 11.6%. ^4^ NE_L_ was a calculated value, while nutrient composition was a measured value.

**Table 2 animals-15-00173-t002:** Effects of four experimental diets on dry matter intake and average daily gain.

Items	Treatment ^1^				SEM	*p*-Value
	LSF	MSF	HSF	ZF		
Dry matter intake, kg/d	8.0	7.7	7.9	8.3		
Average daily gain of body weight, g/d	572.5	633.6	565.7	591.6	35.4	0.349

^1^ LSF = fermented total mixed ration containing 21% rice straw, MSF = fermented total mixed ration containing 28% rice straw, HSF = fermented total mixed ration containing 35% rice straw, ZF = fermented total mixed ration containing 31% sheath and leaves of *Zizania latifolia.*

## Data Availability

Data are available from the corresponding author upon reasonable request.

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
