# Peer review of "Integrating Metabolomics and Genomics to Uncover the Impact of Fermented Total Mixed Ration on Heifer Growth Performance Through Host-Dependent Metabolic Pathways"

_animals, 2025, doi:10.3390/ani15020173_

Round 1

Reviewer 1 Report

Comments and Suggestions for Authors

The paper numbered by 3406557, titled “Integrating metabolomics and genomics to uncover the impact of fermented total mixed ration on heifer growth performance through host-dependent metabolic pathways”, indicated that feeding HSF and ZF FTMR enhances glutathione metabolism and antioxidant capacity in heifers, positioning key metabolites as candidates for ADG markers. However, there are some issues need to modification.

1. The paper does not mention what the basis for selecting 75 of the 157 cattle was, and it is necessary to explain it in the text

2. The ZF group is not very related to other groups, and please explain why this group was added.

3. The conclusion should be rigorous and concise, and this article does not verify it further.

4. Common feeding management does not require citation of references in L115, in addition, check whether the literature and content correspond one-to-one.

5. What is birth weight data used for? There is no explanation in L139.

6. There was no difference in ADG in this experiment, and it is necessary to consider whether it makes sense to analyze the metabolites associated with ADG.

Reviewer 2 Report

Comments and Suggestions for Authors

Thank you for the opportunity to review this interesting article. I appreciate the unique form of data presentation in the manuscript. For my part, I have only a few questions for the authors' consideration:

1.       LINES 123 and 242: the title of the table should not end with a dot (a table is a continuation)

2.       Why are the SEM and P values missing from Table 2 for the parameter Dry matter intake?

3.       What tests were used to verify the normal distribution and homogeneity of variance? Did all the data meet these principles? If not, have non-parametric tests been used? If so, which ones?

4.       The text lacks reference to economic aspects. The addition of such information would enhance the value of this work.

5.       How can matabolomics affect animal production? This information should be detailed in the text.

6.       How can the information in the text influence the knowledge of farmers of, for example, cattle? Such a reference in the manuscript also seems justified.

Reviewer 3 Report

Comments and Suggestions for Authors

In the manuscript, the authors studied the Integrating metabolomics and genomics to uncover the impact of fermented total mixed ration on heifer growth performance through host-dependent metabolic pathways.

However, the following comments can be made.

1. Abstract, introduction. There is no objective of the study.

2. Introduction. Based on the topic of the manuscript, the objective of the study is not clear. Hasn't the feeding of these products to cattle been studied before? It is necessary to correctly identify the objective and relevance of the study of the manuscript.

3. Figure 3. Increase the font size, as it is impossible to read.

4. Materials and Methods. What is the reason for choosing these metabolites? How are they related to metabolic processes and productivity?
